# Bowtie Networks: Generative Modeling for Joint Few-Shot Recognition and Novel-View Synthesis

**Zhipeng Bao**[1]    **Yu-Xiong Wang**[2]    **Martial Hebert**[1]
[1]Carnegie Mellon University    [2]University of Illinois at Urbana-Champaign
{zbao, hebert}@cs.cmu.edu    yxw@illinois.edu

## Abstract

We propose a novel task of joint few-shot recognition and novel-view synthesis: given only one or few images of a novel object from arbitrary views with only category annotation, we aim to simultaneously learn an object classifier and generate images of that type of object from new viewpoints. While existing work copes with two or more tasks mainly by multi-task learning of shareable feature representations, we take a different perspective. We focus on the *interaction and cooperation between a generative model and a discriminative model*, in a way that facilitates knowledge to flow across tasks in complementary directions. To this end, we propose *bowtie networks* that jointly learn 3D geometric and semantic representations *with a feedback loop*. Experimental evaluation on challenging fine-grained recognition datasets demonstrates that our synthesized images are realistic from multiple viewpoints and significantly improve recognition performance as ways of data augmentation, *especially in the low-data regime*. Code and pre-trained models are released at https://github.com/zpbao/bowtie_networks.

## 1 Introduction

Given a never-before-seen object (*e.g.*, a gadwall in Figure 1), humans are able to generalize even from a single image of this object in different ways, including recognizing new object instances and imagining what the object would look like from different viewpoints. Achieving similar levels of generalization for machines is a fundamental problem in computer vision, and has been actively explored in areas such as few-shot object recognition (Fei-Fei et al., 2006; Vinyals et al., 2016; Wang & Hebert, 2016; Finn et al., 2017; Snell et al., 2017) and novel-view synthesis (Park et al., 2017; Nguyen-Phuoc et al., 2018; Sitzmann et al., 2019). However, such exploration is often limited in *separate* areas with specialized algorithms *but not jointly*.

We argue that synthesizing images and recognizing them are inherently interconnected with each other. Being able to *simultaneously* address both tasks with a *single* model is a crucial step toward human-level generalization. This requires learning a richer, shareable internal representation for more comprehensive object understanding than it could be within individual tasks. Such "cross-task" knowledge becomes particularly critical in the low-data regime, where identifying 3D geometric structures of input images facilities recognizing their semantic categories, and vice versa.

Inspired by this insight, here we propose a novel task of *joint few-shot recognition and novel-view synthesis*: given only one or few images of a novel object *from arbitrary views with only category annotation*, we aim to simultaneously learn an object classifier and generate images of that type of object from new viewpoints. This joint task is challenging, because of its (i) *weak supervision*, where we do not have access to any 3D supervision, and (ii) *few-shot setting*, where we need to effectively learn both 3D geometric and semantic representations from minimal data.

While existing work copes with two or more tasks mainly by multi-task learning or meta-learning of a shared feature representation (Yu et al., 2020; Zamir et al., 2018; Lake et al., 2015), we take a different perspective in this paper. Motivated by the nature of our problem, we focus on the *interaction and cooperation between a generative model (for view synthesis) and a discriminative model (for recognition)*, in a way that facilitates knowledge to flow across tasks *in complementary directions*, thus making the tasks help each other. For example, the synthesized images produced by

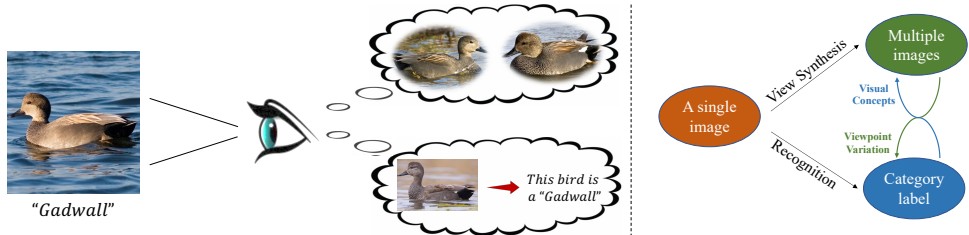

"Gadwall"

Figure 1: **Left:** Given a single image of a novel visual concept (*e.g.*, a gadwall), a person can generalize in various ways, including imagining what this gadwall would look like from different viewpoints (top) and recognizing new gadwall instances (bottom). **Right:** Inspired by this, we introduce a general feedback-based bowtie network that facilitates the interaction and cooperation between a generative module and a discriminative module, thus simultaneously addressing few-shot recognition and novel-view synthesis in the low-data regime.

the generative model provide viewpoint variations and could be used as additional training data to build a better recognition model; meanwhile, the recognition model ensures the preservation of the desired category information and deals with partial occlusions during the synthesis.

To this end, we propose a *feedback-based bowtie network (FBNet)*, as illustrated in Figure 1. The network consists of a view synthesis module and a recognition module, which are linked through feedback connections in a bowtie fashion. This is a general architecture that can be used on top of any view synthesis model and any recognition model. The view synthesis module explicitly learns a 3D geometric representation from 2D images, which is transformed to target viewpoints, projected to 2D features, and rendered to generate images. The recognition module then leverages these synthesized images from different views together with the original real images to learn a semantic feature representation and produce corresponding classifiers, leading to *the feedback from the output of the view synthesis module to the input of the recognition module*. The semantic features of real images extracted from the recognition module are further fed into the view synthesis module as conditional inputs, leading to *the feedback from the output of the recognition module to the input of the view synthesis module*.

One potential difficulty, when combining the view synthesis and the recognition modules, lies in the mismatch in their level of image resolutions. Deep recognition models can benefit from high-resolution images, and the recognition performance greatly improves with increased resolution (Wang et al., 2016; Cai et al., 2019; He et al., 2016). By contrast, it is still challenging for modern generative models to synthesize very high-resolution images (Regmi & Borji, 2018; Nguyen-Phuoc et al., 2019). To address this challenge, while operating on a resolution consistent with state-of-the-art view synthesis models (Nguyen-Phuoc et al., 2019), we further introduce *resolution distillation* to leverage additional knowledge in a recognition model that is learned from higher-resolution images.

**Our contributions** are three-folds. (1) We introduce a new problem of simultaneous few-shot recognition and novel-view synthesis, and address it from a novel perspective of *cooperating generative and discriminative modeling*. (2) We propose feedback-based bowtie networks that jointly learn 3D geometric and semantic representations with feedback in the loop. We further address the mismatch issue between different modules by leveraging resolution distillation. (3) Our approach significantly improves both view synthesis and recognition performance, *especially in the low-data regime*, by enabling direct manipulation of view, shape, appearance, and semantics in generative image modeling.

## 2 RELATED WORK

**Few-Shot Recognition** is a classic problem in computer vision (Thrun, 1996; Fei-Fei et al., 2006). Many algorithms have been proposed to address this problem (Vinyals et al., 2016; Wang & Hebert, 2016; Finn et al., 2017; Snell et al., 2017), including the recent efforts on leveraging generative models (Li et al., 2015; Wang et al., 2018; Schwartz et al., 2018; Zhang et al., 2018; Tsutsui et al., 2019; Chen et al., 2019b; Li et al., 2019; Zhang et al., 2019; Sun et al., 2019). A hallucinator is introduced to generate additional examples in a pre-trained feature space as data augmentation to help with low-shot classification (Wang et al., 2018). MetaGAN improves few-shot recognition by producing fake images as a new category (Zhang et al., 2018). However, these methods either do not synthesize images directly or use a pre-trained generative model that is not optimized towards the downstream task. By contrast, our approach performs joint training of recognition and view synthesis, and enables the two tasks to cooperate through feedback connections. In addition, while there has been work considering both classification and exemplar generation in the few-shot regime, such investigation focuses on simple domains like handwritten characters (Lake et al., 2015) but we address more realistic scenarios with natural images. Note that *our effort is largely orthogonal to*

*designing the best few-shot recognition or novel-view synthesis method*; instead, we show that the joint model outperforms the original methods addressing each task in isolation.

**Novel-View Synthesis** aims to generate a target image with an arbitrary camera pose from one given source image (Tucker & Snavely, 2020). It is also known as "multiview synthesis." For this task, some approaches are able to synthesize lifelike images (Park et al., 2017; Yin & Shi, 2018; Nguyen-Phuoc et al., 2018; Sitzmann et al., 2019; Iqbal et al., 2020; Yoon et al., 2020; Wiles et al., 2020; Wortsman et al., 2020). However, they heavily rely on pose supervision or 3D annotation, which is not applicable in our case. An alternative way is to learn a view synthesis model in an unsupervised manner. Pix2Shape learns an implicit 3D scene representation by generating a 2.5D surfel based reconstruction (Rajeswar et al., 2020). HoloGAN proposes an unsupervised approach to learn 3D feature representations and render 2D images accordingly (Nguyen-Phuoc et al., 2019). Nguyen-Phuoc et al. (2020) learn scene representations from 2D unlabeled images through foreground-background fragmenting. Different from them, not only can our view synthesis module learn from weakly labeled images, but it also enables conditional synthesis to facilitate recognition.

**Feedback-Based Architectures**, where the full or partial output of a system is routed back into the input as part of an iterative cause-and-effect process (Ford, 1999), have been recently introduced into neural networks (Belagiannis & Zisserman, 2017; Zamir et al., 2017; Yang et al., 2018). Compared with prior work, our FBNet contains two complete sub-networks, and the output of *each* module is fed into the other as one of the inputs. Therefore, FBNet is essentially a *bi-directional* feedback-based framework which optimizes the two sub-networks jointly.

**Multi-task Learning** focuses on optimizing a collection of tasks jointly (Misra et al., 2016; Ruder, 2017; Kendall et al., 2018; Pal & Balasubramanian, 2019; Xiao & Marlet, 2020). Task relationships have also been studied (Zamir et al., 2018; Standley et al., 2020). Some recent work investigates the connection between recognition and view synthesis, and makes some attempt to combine them together (Sun et al., 2018; Wang et al., 2018; Xian et al., 2019; Santurkar et al., 2019; Xiong et al., 2020; Michalkiewicz et al., 2020). For example, Xiong et al. (2020) use multiview images to tackle fine-grained recognition tasks. However, their method needs strong pose supervision to train the view synthesis model, while we do not. Also, these approaches do not treat the two tasks of equal importance, *i.e.*, one task as an auxiliary task to facilitate the other. On the contrary, our approach targets the joint learning of the two tasks and improves both of their performance. *Importantly*, we focus on learning a shared generative model, rather than a shared feature representation as is normally the case in multi-task learning.

**Joint Data Augmentation and Task Model Learning** leverage generative networks to improve other visual tasks (Peng et al., 2018; Hu et al., 2019; Luo et al., 2020; Zhang et al., 2020). A generative network and a discriminative pose estimation network are trained jointly through adversarial loss in Peng et al. (2018), where the generative network performs data augmentation to facilitate the downstream pose estimation task. Luo et al. (2020) design a controllable data augmentation method for robust text recognition, which is achieved by tracking and refining the moving state of the control points. Zhang et al. (2020) study and make use of the relationship among facial expression recognition, face alignment, and face synthesis to improve training. Mustikovela et al. (2020) leverage a generative model to boost viewpoint estimation. The main difference is that we focus on the joint task of synthesis and recognition and achieve bi-directional feedback, while existing work only considers optimizing the target discriminative task using adversarial training or with a feedforward network.

## 3 OUR APPROACH

### 3.1 JOINT TASK OF FEW-SHOT RECOGNITION AND NOVEL-VIEW SYNTHESIS

**Problem Formulation:** Given a dataset $\mathcal{D} = \{(x_i, y_i)\}$, where $x_i \in \mathcal{X}$ is an image of an object and $y_i \in \mathcal{C}$ is the corresponding category label ($\mathcal{X}$ and $\mathcal{C}$ are the image space and label space, respectively), we address the following two tasks *simultaneously*. (i) Object recognition: learning a discriminative model $R : \mathcal{X} \to \mathcal{C}$ that takes as input an image $x_i$ and predicts its category label. (ii) Novel-view synthesis: learning a generative model $G : \mathcal{X} \times \Theta \to \mathcal{X}$ that, given an image $x_i$ of category $y_i$ and an arbitrary 3D viewpoint $\theta_j \in \Theta$, synthesizes an image in category $y_i$ viewed from $\theta_j$. Notice that we are more interested in *category-level consistency*, for which $G$ is able to generate images of not only the instance $x_i$ but also other objects of the category $y_i$ from different viewpoints. This joint-task scenario requires us to improve the performance of both 2D and 3D tasks under weak supervision *without any ground-truth 3D annotations*. Hence, we need to exploit the *cooperation* between them.

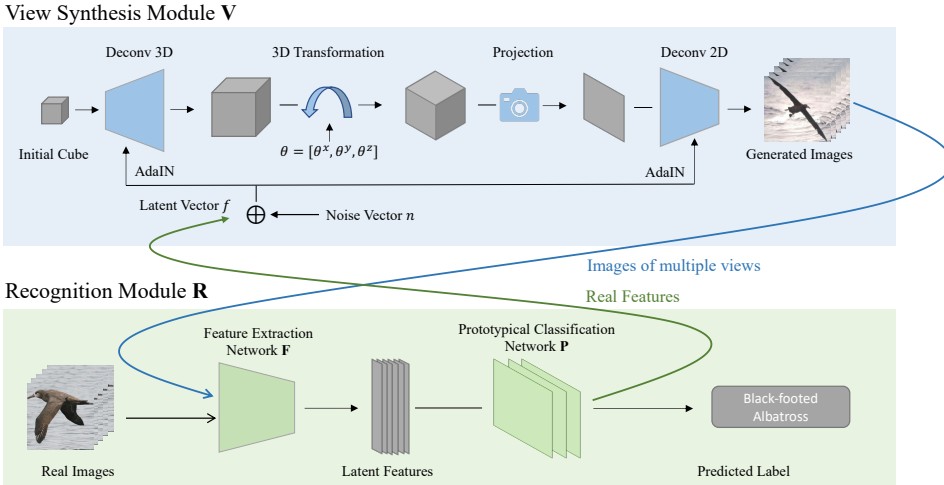

Figure 2: Architecture of our feedback-based bowtie network. The whole network consists of a view synthesis module and a recognition module, which are linked through feedback connections in a bowtie fashion.

**Few-Shot Setting:** The few-shot dataset consists of one or only a few images per category, which makes our problem even more challenging. To this end, following the recent work on knowledge transfer and few-shot learning (Hariharan & Girshick, 2017; Chen et al., 2019a), we leverage a set of "base" classes $\mathcal{C}_{\text{base}}$ with a large-sample dataset $\mathcal{D}_{\text{base}} = \{(x_i, y_i), y_i \in \mathcal{C}_{\text{base}}\}$ to train our initial model. We then fine-tune the pre-trained model on our target "novel" classes $\mathcal{C}_{\text{novel}}$ ($\mathcal{C}_{\text{base}} \cap \mathcal{C}_{\text{novel}} = 0$) with its small-sample dataset $\mathcal{D}_{\text{novel}} = \{(x_i, y_i), y_i \in \mathcal{C}_{\text{novel}}\}$ (*e.g.*, a $K$-shot setting corresponds to $K$ images per class).

## 3.2 FEEDBACK-BASED BOWTIE NETWORKS

To address the joint task, we are interested in learning a generative model that can synthesize realistic images of different viewpoints, which are also useful for building a strong recognition model. We propose a feedback-based bowtie network (FBNet) for this purpose. This model consists of a view synthesis module and a recognition module, trained in a joint, end-to-end fashion. Our key insight is to explicitly introduce feedback connections between the two modules, so that they cooperate with each other, thus enabling the entire model to simultaneously learn 3D geometric and semantic representations. This general architecture can be used on top of any view synthesis model and any recognition model. Here we focus on a state-of-the-art view synthesis model – HoloGAN (Nguyen-Phuoc et al., 2019), and a widely adopted few-shot recognition model – prototypical network (Snell et al., 2017), as shown in Figure 2.

### 3.2.1 VIEW SYNTHESIS MODULE

The view synthesis module $V$ is shown in the blue shaded region in Figure 2. It is adapted from HoloGAN (Nguyen-Phuoc et al., 2019), a state-of-the-art model for unsupervised view synthesis. This module consists of a generator $G$ which first generates a 3D feature representation from a latent constant tensor (initial cube) through 3D convolutions. The feature representation is then transformed to a certain pose and projected to 2D with a projector. The final color image is then computed through 2D convolutions. This module takes two inputs: a latent vector input $z$ and a view input $\theta$. $z$ characterizes the style of the generated image through adaptive instance normalization (AdaIN) (Huang & Belongie, 2017) units. $\theta = [\theta^x, \theta^y, \theta^z]$ guides the transformation of the 3D feature representation. This module also contains a discriminator $D$ to detect whether an image is real or fake (not shown in Figure 2). We use the standard GAN loss from DC-GAN (Radford et al., 2016), $\mathcal{L}_{\text{GAN}}(G, D)$. We make the following important modifications to make the architecture applicable to our joint task.

**Latent Vector Formulation:** To allow the synthesis module to get feedback from the recognition module (details are shown in Section 3.2.3), we first change HoloGAN from unconditional to conditional. To this end, we model the latent input $z$ as: $z_i = f_i \oplus n_i$, where $f_i$ is the conditional feature input derived from image $x_i$ and $n_i$ is a noise vector sampled from Gaussian distribution. $\oplus$ is the combination strategy (*e.g.*, concatenation). By doing so, the synthesis module leverages

additional semantic information, and thus maintains the category-level consistency with a target image and improves the diversity of the generated images.

**Identity Regularizer:** Inspired by Chen et al. (2016), we introduce an identity regularizer to ensure that the synthesis module simultaneously satisfies two critical properties: (i) the identity of the generated image remains when we only change the view input $\theta$; (ii) the orientation of the generated image preserves when we only change the latent input $z$, and this orientation should be consistent with the view input $\theta$. Specifically, we leverage an encoding network $H$ to predict the reconstructed latent vector $z'$ and the view input $\theta'$: $H(G(z, \theta)) = [z', \theta']$, where $G(z, \theta)$ is the generated image. Then we minimize the difference between the real and the reconstructed inputs as

$$\mathcal{L}_{\text{identity}}(G, H) = \mathbb{E}_z \|z - z'\|^2 + \mathbb{E}_\theta \|\theta - \theta'\|^2. \tag{1}$$

Here $H$ shares the majority of the convolution layers of the discriminator $D$, but uses an additional fully-connected layer. Section A explains the detailed architecture of the view synthesis module.

### 3.2.2 RECOGNITION MODULE

The recognition module $R$ (green shaded region in Fig. 2) consists of a feature extraction network $F$ which transforms images to latent features, and a prototypical classification network $P$ (Snell et al., 2017) which performs the final classification. Below we explain the design of these two components, focusing on how to address the technical challenges faced by joint training with view synthesis.

**Feature Extraction with Resolution Distillation:** We use a ResNet (He et al., 2016) as our feature extraction network $F$ to transform images into latent features for the recognition module. One of the main obstacles to combining $F$ with the synthesis module is that state-of-the-art synthesis models and recognition models operate on different resolutions. Concretely, to the best of our knowledge, current approaches to unsupervised novel-view synthesis still cannot generate satisfactory high-resolution images (*e.g.*, $224 \times 224$) (Nguyen-Phuoc et al., 2019). By contrast, the performance of current well-performing recognition models substantially degrades with low-resolution images (Wang et al., 2016; Cai et al., 2019). To reconcile the resolution incompatibility, we introduce a simple distillation technique inspired by the general concept of knowledge distillation (Hinton et al., 2014). Specifically, we operate on the resolution of the synthesis module (*e.g.*, $64 \times 64$). But we benefit from an additional auxiliary feature extraction network $F_{\text{highR}}$ that is trained on high-resolution images (*e.g.*, $224 \times 224$). We first pre-train $F_{\text{highR}}$ following the standard practice with a cross-entropy softmax classifier (Liu et al., 2016). We then train our feature extraction network $F_{\text{lowR}}$ (the one used in the recognition module), under the guidance of $F_{\text{highR}}$ through matching their features:

$$\mathcal{L}_{\text{feature}}(F_{\text{lowR}}) = \mathbb{E}_x \|F_{\text{highR}}(x) - F_{\text{lowR}}(x)\|^2, \tag{2}$$

where $x$ is a training image. With the help of resolution distillation, the feature extraction network re-captures information in high-resolution images but potentially missed in low-resolution images.

**Prototypical Classification Network:** We use the prototypical network $P$ (Snell et al., 2017) as our classifier. The network assigns class probabilities $\hat{p}$ based on distance of the input feature vector from class centers $\mu$; and $\mu$ is calculated by using support images in the latent feature space:

$$\hat{p}_c(x) = \frac{e^{-d(P(F_{\text{lowR}}(x)), \mu_c)}}{\sum_j e^{-d(P(F_{\text{lowR}}(x)), \mu_j)}}, \quad \mu_c = \frac{\sum_{(x_i, y_i) \in S} P(F_{\text{lowR}}(x_i)) \mathbf{I}[y_i = c]}{\sum_{(x_i, y_i) \in S} \mathbf{I}[y_i = c]}, \tag{3}$$

where $x$ is a real query image, $\hat{p}_c$ is the probability of category $c$, and $d$ is a distance metric (*e.g.*, Euclidean distance). $S$ is the support dataset. $P$ operates on top of the feature extraction network $F$, and consists of 3 fully-connected layers as additional feature embedding (the classifier is non-parametric). Another benefit of using the prototypical network lies in that it enables the recognition module to explicitly leverage the generated images in a way of data augmentation, *i.e.*, $S$ contains both real and generated images to compute the class mean. Notice that, though, the module parameters are updated based on the loss calculated on the *real query images*, which is a cross-entropy loss $\mathcal{L}_{\text{rec}}(R)$ between their predictions $\hat{p}$ and ground-truth labels.

### 3.2.3 Feedback-Based Bowtie Model

As shown in Figure 2, we leverage a bowtie architecture for our full model, where the output of each module is fed into the other module as one of its inputs. Through joint training, such connections work as explicit feedback to facilitate the communication and cooperation between different modules.

**Feedback Connections:** We introduce two complementary feedback connections between the view synthesis module and the recognition module: (1) **recognition output → synthesis input** (green arrow in Figure 2), where the features of the real images extracted from the recognition module are fed into the synthesis module as conditional inputs to generate images from different views; (2) **synthesis output → recognition input** (blue arrow in Figure 2), where the generated images are used to produce an augmented set to train the recognition module.

**Categorical Loss for Feedback:** The view synthesis module needs to capture the categorical semantics in order to further encourage the generated images to benefit the recognition. Therefore, we introduce a categorical loss to update the synthesis module with the prediction results of the generated images:

$$\mathcal{L}_{\text{cat}}(G) = \mathbb{E}_{y_i} \| - \log(R(G(z_i, \theta_i))) \|,$$ (4)

where $y_i$ is the category label for the generated image $G(z_i, \theta_i)$. This loss also implicitly increases the diversity and quality of the generated images.

**Final Loss Function:** The final loss function is:

$$\mathcal{L}_{\text{Total}} = \mathcal{L}_{\text{GAN}} + \mathcal{L}_{\text{rec}} + \mathcal{L}_{\text{feature}} + \lambda_{\text{id}}\mathcal{L}_{\text{identity}} + \lambda_{\text{cat}}\mathcal{L}_{\text{cat}},$$ (5)

where $\lambda_{\text{id}}$ and $\lambda_{\text{cat}}$ are trade-off hyper-parameters.

**Training Procedure:** We first pre-train $F_{\text{highR}}$ on the high-resolution dataset and save the computed features. These features are used to help train the feature extraction network $F_{\text{lowR}}$ through $\mathcal{L}_{\text{feature}}$. Then the entire model is first trained on $\mathcal{C}_{\text{base}}$ and then fine-tuned on $\mathcal{C}_{\text{novel}}$. The training on the two sets are similar. During each iteration, we randomly sample some images per class as a support set and one image per class as a query set. The images in the support set, together with their computed features via the entire recognition module, are fed into the view synthesis module to generate multiple images from different viewpoints. These synthesized images are used to augment the original support set to compute the prototypes. Then, the query images are used to update the parameters of the recognition module through $\mathcal{L}_{\text{rec}}$; the view-synthesis module is updated through $\mathcal{L}_{\text{GAN}}$, $\mathcal{L}_{\text{identity}}$, and $\mathcal{L}_{\text{cat}}$. The entire model is trained in an end-to-end fashion. More details are in Section B.

## 4 Experimental Evaluation

**Datasets:** We focus on two datasets here: the Caltech-UCSD Birds (CUB) dataset which contains 200 classes with 11,788 images (Welinder et al., 2010), and the CompCars dataset which contains 360 classes with 25,519 images (Yang et al., 2015). Please refer to Section C for more details of the datasets. These are challenging fine-grained recognition datasets for our joint task. The images are resized to $64 \times 64$. We randomly split the entire dataset into 75% as the training set and 25% as the test set. For CUB, 150 classes are selected as base classes and 50 as novel classes. For CompCars, 240 classes are selected as base classes and 120 as novel classes. Note that we focus on simultaneous recognition and synthesis over *all* base or novel classes, which is significantly more challenging than typical 5-way classification over sampled classes in most of few-shot classification work (Snell et al., 2017; Chen et al., 2019a). We also include evaluation on additional datasets in Section D.

**Implementation Details:** We set $\lambda_{\text{id}} = 10$ and $\lambda_{\text{cat}} = 1$ via cross-validation. We use ResNet-18 (He et al., 2016) as the feature extraction network, unless otherwise specified. To match the resolution of our data, we change the kernel size of the first convolution layer of ResNet from 7 to 5. The training process requires hundreds of examples at each iteration, which may not fit in the memory of our device. Hence, inspired by Wang et al. (2018), we make a trade-off to first train the feature extraction network through resolution distillation. We then freeze its parameters and train the other parts of our model. Section C includes more implementation details.

**Compared Methods:** Our feedback connections enable the two modules to cooperate through joint training. Therefore, to evaluate the effectiveness of the feedback connections, we focus on the following comparisons. (1) For the novel-view image synthesis task, we compare our approach **FBNet** with the state-of-the-art method **HoloGAN** (Nguyen-Phuoc et al., 2019). We also consider a

| | Model | Base | Novel-$K$=1 | Novel-$K$=5 |
|---|---|---|---|---|
| CUB | FBNet-rec | 57.91 | $47.53 \pm 0.14$ | $71.26 \pm 0.26$ |
| | FBNet-aug | 58.03 | $47.20 \pm 0.19$ | $71.51 \pm 0.33$ |
| | FBNet | **59.43** | $\mathbf{48.39 \pm 0.19}$ | $\mathbf{72.76 \pm 0.24}$ |
| CompCars | FBNet-rec | 46.05 | $20.83 \pm 0.03$ | $50.52 \pm 0.11$ |
| | FBNet-aug | 47.41 | $21.59 \pm 0.05$ | $51.07 \pm 0.14$ |
| | FBNet | **49.63** | $\mathbf{23.28 \pm 0.05}$ | $\mathbf{53.12 \pm 0.09}$ |

Table 1: Top-1 (%) recognition accuracy on the CUB and CompCars datasets. For base classes: **150-way** classification on CUB and **240-way** classification on CompCars; for $K$-shot novel classes: **50-way** classification on CUB and **120-way** classification on CompCars. Our FBNet consistently achieves the best performance for both base and novel classes, and joint training significantly outperforms training each module individually.

| | Model | IS (↑) | | | FID (↓) | | |
|---|---|---|---|---|---|---|---|
| | | Base | Novel-$K$=1 | Novel-$K$=5 | Base | Novel-$K$=1 | Novel-$K$=5 |
| CUB | *Real Images* | $4.55 \pm 0.30$ | $3.53 \pm 0.22$ | $3.53 \pm 0.22$ | 0 | 0 | 0 |
| | HoloGAN (Nguyen-Phuoc et al., 2019) | $3.55 \pm 0.09$ | $2.44 \pm 0.07$ | $2.58 \pm 0.08$ | 79.01 | 106.56 | 94.73 |
| | FBNet-view | $3.60 \pm 0.12$ | $2.53 \pm 0.03$ | $2.64 \pm 0.05$ | 75.38 | 107.36 | 103.25 |
| | FBNet | $\mathbf{3.69 \pm 0.17}$ | $\mathbf{2.79 \pm 0.06}$ | $\mathbf{2.83 \pm 0.12}$ | **70.86** | **104.04** | **92.97** |
| CompCars | *Real Images* | $2.96 \pm 0.12$ | $2.80 \pm 0.13$ | $2.80 \pm 0.13$ | 0 | 0 | 0 |
| | HoloGAN (Nguyen-Phuoc et al., 2019) | $1.85 \pm 0.08$ | $1.41 \pm 0.04$ | $1.65 \pm 0.07$ | 51.49 | 93.48 | 83.17 |
| | FBNet-view | $2.03 \pm 0.09$ | $1.44 \pm 0.05$ | $1.71 \pm 0.07$ | 49.94 | 92.01 | 83.58 |
| | FBNet | $\mathbf{2.33 \pm 0.14}$ | $\mathbf{1.89 \pm 0.07}$ | $\mathbf{1.91 \pm 0.10}$ | **44.70** | **89.39** | **78.38** |

Table 2: Novel-view synthesis results under the FID and IS metrics. ↑ indicates that higher is better, and ↓ indicates that lower is better. As a reference, FID and IS of *Real Images* represent the best results we could expect. FBNet consistently outperforms the baselines, achieving 18% improvements for FID and 19% for IS.

variant of our approach **FBNet-view**, which has the same architecture as our novel-view synthesis module, but takes the *constant* features extracted by a pre-trained ResNet-18 as latent input. FBNet-view can be also viewed as a *conditional* version of HoloGAN. (2) For the few-shot recognition task, we compare our full model **FBNet** with its two variants: **FBNet-rec** inherits the architecture of our recognition module, which is essentially a prototypical network (Snell et al., 2017); **FBNet-aug** uses the synthesized images from *individually trained* FBNet-view as data augmentation for the recognition module. Note that, while conducting comparisons with other few-shot recognition (*e.g.*, Chen et al. (2019a); Finn et al. (2017)) or view synthesis models (*e.g.*, Yoon et al. (2020); Wiles et al. (2020)) is interesting, *it is not the main focus of this paper*. We aim to validate that the feedback-based bowtie architecture outperforms the single-task models upon which it builds, rather than designing the best few-shot recognition or novel-view synthesis method. In Section F, we show that our framework is general and can be used on top of other single-task models and improve their performance. All the models are trained following the same few-shot setting described in Section 3.1.

**View Synthesis Facilitates Recognition:** Table 1 presents the top-1 recognition accuracy for the base classes and the novel classes, respectively. We focus on the challenging 1, 5-shot settings, where the number of training examples per novel class $K$ is 1 or 5. For the novel classes, we run five trials for each setting of $K$, and report the average accuracy and standard deviation for all the approaches. Table 1 shows that our FBNet *consistently* achieves the best few-shot recognition performance on the two datasets. Moreover, the significant improvement of FBNet over FBNet-aug (where the recognition model uses additional data from the *conditional* view synthesis model, but they are trained separately) indicates that the feedback-based *joint training* is the key to improve the recognition performance.

**Recognition Facilitates View Synthesis:** We investigate the novel-view synthesis results under two standard metrics. The **FID** score computes the Fréchet distance between two Gaussians fitted to feature representations of the source (real) images and the target (synthesized) images (Dowson & Landau, 1982). The **Inception Score (IS)** uses an Inception network pre-trained on ImageNet (Deng et al., 2009) to predict the label of the generated image and calculate the entropy based on the predictions. IS seeks to capture both the quality and diversity of a collection of generated images (Salimans et al., 2016). A higher IS or a lower FID value indicates better realism of the generated images. A larger variance of IS indicates more diversity of the generated images. We generate images of random views in one-to-one correspondence with the training examples for all the models, and compute the IS and FID values based on these images. The results are reported in Table 2. As a reference, we also show the results of *real images* under the two metrics, which are the best results we could expect from synthesized images. Our FBNet consistently achieves the best performance under both metrics. Compared with HoloGAN, our method brings up to 18% improvement under FID and 19% under IS. Again, the significant performance gap between FBNet and FBNet-view shows that the feedback-based *joint training* substantially improves the synthesis performance.

IS and FID cannot effectively evaluate whether the generated images maintain the category-level identity and capture different viewpoints. Therefore, Figure 3 visualizes the synthesized multiview images. Note that, in our problem setting of limited training data under weak supervision, we could

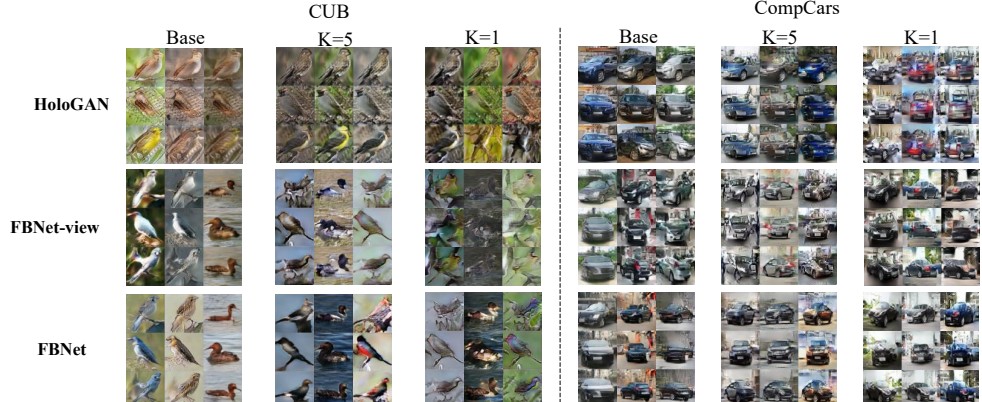

Figure 3: Synthesized images from multiple viewpoints. Images in the same row/column are from the same viewpoint/object. Our approach captures the shape and attributes well *even in the extremely low-data regime*.

| Setting | Model | ResNet-10 | ResNet-18 | ResNet-34 | ResNet-50 |
|---------|-------|-----------|-----------|-----------|-----------|
| $K$=1 | FBNet-view | 46.28 | 47.53 | 46.79 | 45.68 |
|  | FBNet | **48.85** | **48.39** | **47.65** | **47.03** |
| $K$=5 | FBNet-view | 71.66 | 71.26 | 70.69 | 70.00 |
|  | FBNet | **72.49** | **72.76** | **71.28** | **70.95** |

Table 3: Few-shot recognition accuracy consistently improves with different feature extraction networks.

| Setting | $K$=1 | | | $K$=5 | | |
|---------|-------|---------|---------|-------|---------|---------|
|  | Acc | FID ($\downarrow$) | IS ($\uparrow$) | Acc | FID ($\downarrow$) | IS ($\uparrow$) |
| Multitask-Feat (Ruder, 2017) | 34.71 | 110.03 | $2.19 \pm 0.03$ | 52.54 | 99.61 | $2.44 \pm 0.04$ |
| FBNet w/o Dist | 22.47 | 108.73 | $2.31 \pm 0.05$ | 34.15 | 97.64 | $2.42 \pm 0.07$ |
| FBNet w/o Proto | 44.62 | 105.81 | $2.61 \pm 0.07$ | 70.04 | 95.15 | $2.76 \pm 0.10$ |
| FBNet | **48.39** | **104.0**4 | **$2.79 \pm 0.06$** | **72.76** | **92.97** | **$2.83 \pm 0.12$** |

Table 4: Ablation studies on CUB regarding (i) learning a shared feature representation through standard multi-task learning, (ii) FBNet without resolution distillation, and (iii) FBNet using a regular classification network without prototypical classification. Our full model achieves the best performance.

not expect that the quality of the synthesized images would match those generated based on large amounts of training data, *e.g.* Brock et al. (2019). This demonstrates the general difficulty of image generation in the few-shot setting, which is worth further exploration in the community.

Notably, even in this challenging setting, our synthesized images are of significantly higher visual quality than the state-of-the-art baselines. Specifically, (1) our FBNet is able to perform *controllable* conditional generation, while HoloGAN cannot. Such conditional generation enables FBNet to better capture the shape information of different car models on CompCars, which is crucial to the recognition task. On CUB, FBNet captures both the shape and attributes well even in the extremely low-data regime (1-shot), thus generating images of higher quality and more diversity. (2) Our FBNet also better maintains the identity of the objects in different viewpoints. For both HoloGAN and FBNet-view, it is hard to tell whether they keep the identity, but FBNet synthesizes images well from all the viewpoints while maintaining the main color and shape. (3) In addition, we notice that there is just a minor improvement for the visual quality of the synthesis results from HoloGAN to FBNet-view, indicating that simply changing the view synthesis model from unconditional to conditional versions does not improve the performance. However, through our feedback-based joint training with recognition, the quality and diversity of the generated images significantly improve.

**Shared Generative Model vs. Shared Feature Representation:** We further compare with a standard multi-task baseline (Ruder, 2017), which learns a shared feature representation across the joint tasks, denoted as 'Multitask-Feat' in Table 4. We treat the feature extraction network as a shared component between the recognition module and the view synthesis module, and update its parameters using both tasks *without* feedback connections. Table 4 shows that, through the feedback connections, our shared generative model captures the *underlying image generation mechanism* for more comprehensive object understanding, outperforming direct task-level shared feature representation.

**Ablation – Different Recognition Networks:** While we used ResNet-18 as the default feature extraction network, our approach is applicable to different recognition models. Table 3 shows that the recognition performance with different feature extraction networks consistently improves. Interestingly, ResNet-10/18 outperform the deeper models, indicating that the deeper models might suffer from over-fitting in few-shot regimes, consistent with the observation in Chen et al. (2019a).

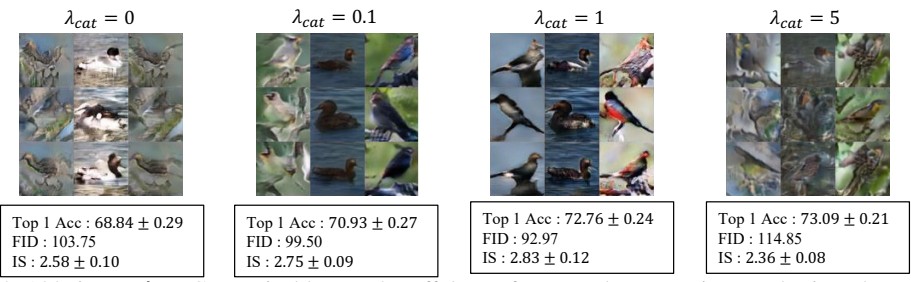

Figure 4: Ablation on $\lambda_{\text{cat}}$. Categorical loss trades off the performance between view synthesis and recognition.

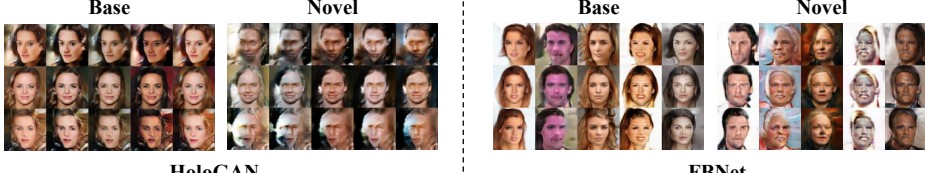

Figure 5: Synthesized images by HoloGAN and FBNet on CelebA-HQ. Few-shot attributes (left to right): Black Hair, Gray Hair, Bald, Wearing Hat, and Aging. FBNet synthesizes images of higher quality and diversity.

**Ablation – Categorical Loss:** In addition to the feedback connections, our synthesis and recognition modules are linked by the categorical loss. To analyze its effect, we vary $\lambda_{\text{cat}}$ among 0 (without the categorical loss), 0.1, 1, and 5. Figure 4 shows the quantitative and qualitative results on CUB. With $\lambda_{\text{cat}}$ increasing, the recognition performance improves gradually. Meanwhile, a too large $\lambda_{\text{cat}}$ reduces the visual quality of the generated images: checkerboard noise appears. While these images are not visually appealing, they still benefit the recognition task. This shows that the categorical loss trades off the performance between the two tasks, and there is a "sweet spot" between them.

**Ablation – Resolution Distillation and Prototypical Classification:** Our proposed resolution distillation reconciles the resolution inconsistency between the synthesis and recognition modules, and further benefits from a recognition model trained on high-resolution images. The prototypical network leverages the synthesized images, which constitutes one of the feedback connections. We evaluate their effect by building two variants of our model without these techniques: 'FBNet w/o Dist' trains the feature extraction network directly from low-resolution images; 'FBNet w/o Proto' uses a regular classification network instead of the prototypical network. Table 4 shows that the performance of full FBNet significantly outperforms these variants, verifying the importance of our techniques.

**Qualitative Results on the CelebA-HQ Dataset:** We further show that the visual quality of our synthesized images significantly gets improved on datasets *with better aligned poses*. For this purpose, we conduct experiments on CelebA-HQ (Lee et al., 2020), which contains 30,000 aligned human face images regarding 40 attributes in total. We randomly select 35 attributes as training attributes and 5 as few-shot test attributes. While CelebA-HQ does not provide pose annotation, the aligned faces mitigate the pose issue to some extent. Figure 5 shows that both the visual quality and diversity of our synthesized images substantially improve, while consistently outperforming HoloGAN.

**Discussion and Future Work:** Our experimental evaluation has focused on fine-grained categories, mainly because state-of-the-art novel-view synthesis models still cannot address image generation for a wide spectrum of general images (Liu et al., 2019). Meanwhile, our feedback-based bowtie architecture is general. With the advance in novel-view synthesis, such as the recent work of BlockGAN (Nguyen-Phuoc et al., 2020) and RGBD-GAN (Noguchi & Harada, 2020), our framework could be potentially extended to deal with broader types of images. Additional further investigation includes exploring more architecture choices and dealing with images with more than one object.

## 5 CONCLUSION

This paper has proposed a feedback-based bowtie network for the joint task of few-shot recognition and novel-view synthesis. Our model consistently improves performance for both tasks, especially with extremely limited data. The proposed framework could be potentially extended to address more tasks, leading to a generative model useful and shareable across a wide range of tasks.

**Acknowledgement:** This work was supported in part by ONR MURI N000014-16-1-2007 and by AFRL Grant FA23861714660. We also thank NVIDIA for donating GPUs and AWS Cloud Credits for Research program.

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

APPENDIX

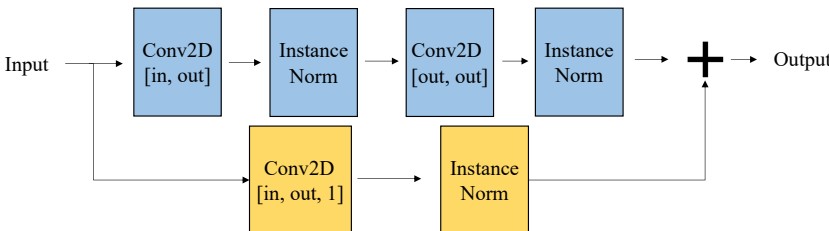

Figure A: Architecture of ResBlock used in the view synthesis module. The default kernel size is 3 and the stride is 1.

## A  DETAILED ARCHITECTURE OF VIEW SYNTHESIS MODULE

One of the central components in our view synthesis module is the ResBlock adapted from ResNet (He et al., 2016), where we use Instance Normalization instead of Batch Normalization (Ulyanov et al., 2016; Ioffe & Szegedy, 2015). Figure A shows the architecture of a 2D ResBlock. For all the convolution layers in this structure, the kernel size is 3 and the stride is 1. By changing the 2D convolution layers to 3D convolution layers, we will get a 3D ResBlock. Figure B shows the architecture of the discriminator $D$. The kernel size is 3 and the stride is 2 for all the convolution layers. The structure of the generator $G$ is illustrated in Figure C. Notice that the "Res-Up" module is the structure of ResBlock, followed by an upsampling layer. The kernel size is 3 and the stride is 1 for all the convolution layers.

## B  PSEUDOCODE OF TRAINING ALGORITHM

To have a better understanding of the training procedure of **FBNet**, we include Algorithm 1 to show the detailed training process on *base* classes. The training on novel classes follows similar process, except the sample number $n$ ($n = 1$ when we conduct 1-shot training).

## C  ADDITIONAL EXPERIMENTAL DETAILS

**Data Pre-processing:** For the CUB (Welinder et al., 2010) dataset, we first square crop the images with the given bounding boxes, and then we resize the cropped images to 64-resolution. For the CompCars (Yang et al., 2015) dataset, slightly different from the original paper and Holo-GAN (Nguyen-Phuoc et al., 2019), we follow the instructions of the publicly released CompCars dataset[1] for classification tasks, and obtain 366 classes of car models after standard pre-processing. We then manually drop 6 classes which have fewer than 15 images per class, and construct a proper dataset for our experiments.

**Additional Implementation Details:** In the main paper, we set $\lambda_{\text{id}} = 10$ and $\lambda_{\text{cat}} = 1$ via cross-validation, and found that the performance is relatively stable to the setting of these trade-off hyper-parameters. We sample 5 images per class for $\mathcal{C}_{\text{base}}$ and 1 image for $\mathcal{C}_{\text{novel}}$. We use Adam optimizer for all the networks. The learning rate is set to $5e - 5$. The final dimension of the feature extraction network is 1,000. The hidden size of all the three fully-connected layers is 128, and the

---

[1] http://mmlab.ie.cuhk.edu.hk/datasets/comp_cars/

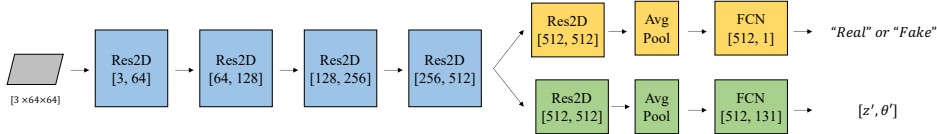

Figure B: Architecture of the discriminator in the view synthesis module. The kernel size of all the convolution layers is 3 and the stride is 2.

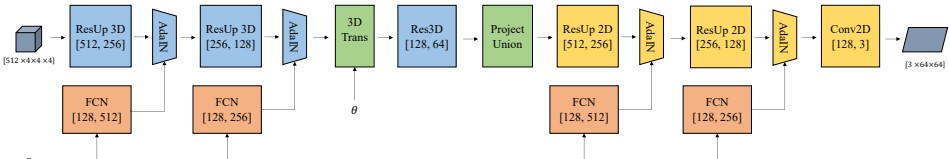

Figure C: Architecture of the generator in the view synthesis module. Res-Up module is a combination of a ResBlock and an upsampling layer.

final feature dimension of the prototypical classification network is also 128. The batch size is 64 for the view synthesis module. We train 1,400 iterations for $\mathcal{C}_{\text{base}}$ and 100 iterations for $\mathcal{C}_{\text{novel}}$.

---

**Algorithm 1:** Training process of FBNet on base classes.

---

**Initialization:**
$max\_it$: Maximum iteration for the training;
$R$: Recognition module, $V$: View synthesis module;
$F$: Feature extraction network, $F_{\text{high}}$: Feature extraction network with high-resolution images;
$G$: Generator of view synthesis module, $D$: Discriminator of view synthesis module;
$n$: Number of support images per class, $n = 5$ ;
**for** $iter \leftarrow 1$ **to** $max\_iter$ **do**
    $S_{\text{support}} = \{\}, S_{\text{query}} = \{\}, S_{\text{augmented}} = \{\}$;
    **for** $c \in \mathcal{C}_{base}$ **do**
        support_ims $\leftarrow$ sample $n$ images in $c$;
        query_ims $\leftarrow$ sample 1 image in $c$;
        $S_{\text{support}} \leftarrow S_{\text{support}} \cup$ support_ims ;
        $S_{\text{query}} \leftarrow S_{\text{query}} \cup$ query_ims ;
    **end**
    $f_{\text{high}} \leftarrow F_{\text{high}}(S_{\text{support}} \cup S_{\text{query}})$;
    $f_{\text{low}} \leftarrow F(S_{\text{support}} \cup S_{\text{query}})$ ;
    **for** $img$ **in** $S_{support}$ **do**
        $f = R(img)$ ;
        $z = f \oplus \mathcal{N}$ ;
        $\theta \leftarrow$ sample a view angle;
        $img' \leftarrow G(z, \theta)$;
        $y = D(img)$;
        $[y', z', \theta'] = D(img')$ ;
        $\mathcal{L}_{\text{GAN}}(y, img, y', img') \rightarrow$ update $G, D$;
        $\mathcal{L}_{\text{id}}(z, \theta, z', \theta') \rightarrow$ update $D$;
        $S_{\text{augmented}} \leftarrow S_{\text{augmented}} \cup img'$ ;
    **end**
    $S_{\text{whole}} \leftarrow S_{\text{support}} \cup S_{\text{augmented}}$ ;
    $\mathcal{L}_{\text{rec}}(S_{\text{whole}}, S_{\text{query}}) \rightarrow$ update $R$;
    $\mathcal{L}_{\text{feature}}(f_{\text{high}}, f_{\text{real}}) \rightarrow$ update $F$;
    $\mathcal{L}_{\text{cat}}(S_{\text{support}}, S_{\text{augmented}}) \rightarrow$ update $G$;
**end**

---

## D EXPERIMENTS ON ADDITIONAL DATASETS

To show the effectiveness and generality of our proposed FBNet, we conduct experiments on two additional datasets: the North American Birds (NAB) dataset (Van Horn et al., 2015) and the Stanford Dog dataset (DOG) (Khosla et al., 2011). There are 555 classes with 48,527 images in NAB dataset.

|     | Model | Base | Novel-$K$=1 | Novel-$K$=5 |
|-----|-------|------|-------------|-------------|
| NAB | FBNet-rec | 44.56 | $23.97 \pm 0.05$ | $58.09 \pm 0.19$ |
|     | FBNet-aug | 44.85 | $23.69 \pm 0.08$ | $58.40 \pm 0.26$ |
|     | FBNet | **45.63** | $\mathbf{24.15 \pm 0.07}$ | $\mathbf{58.98 \pm 0.15}$ |
| DOG | FBNet-rec | 51.13 | $53.33 \pm 0.09$ | $72.59 \pm 0.17$ |
|     | FBNet-aug | 51.46 | $53.11 \pm 0.11$ | $72.68 \pm 0.15$ |
|     | FBNet | **52.25** | $\mathbf{53.78 \pm 0.08}$ | $\mathbf{73.21 \pm 0.14}$ |

Table A: Top-1 (%) recognition accuracy on the NAB and DOG datasets. For base classes: **350-way** classification on NAB and **80-way** classification on DOG; for $K$-shot novel classes: **205-way** classification on NAB and **40-way** classification on DOG. Again, our FBNet consistently achieves the best performance for both base and novel classes.

|     | Model | IS ($\uparrow$) | | | FID ($\downarrow$) | | |
|-----|-------|------|-------------|-------------|------|-------------|-------------|
|     |       | Base | Novel-$K$=1 | Novel-$K$=5 | Base | Novel-$K$=1 | Novel-$K$=5 |
| NAB | *Real Images* | $4.90 \pm 0.31$ | $3.47 \pm 0.14$ | $3.88 \pm 0.19$ | 0 | 0 | 0 |
|     | HoloGAN (Nguyen-Phuoc et al., 2019) | $4.06 \pm 0.08$ | $2.52 \pm 0.04$ | $2.65 \pm 0.06$ | 47.52 | 85.74 | 76.38 |
|     | FBNet | $\mathbf{4.13 \pm 0.12}$ | $\mathbf{2.90 \pm 0.04}$ | $\mathbf{3.05 \pm 0.08}$ | **40.00** | **74.55** | **62.39** |
| DOG | *Real Images* | $8.62 \pm 0.36$ | $4.18 \pm 0.11$ | $4.91 \pm 0.19$ | 0 | 0 | 0 |
|     | HoloGAN (Nguyen-Phuoc et al., 2019) | $6.06 \pm 0.29$ | $3.35 \pm 0.11$ | $3.85 \pm 0.17$ | 53.85 | 82.64 | 73.21 |
|     | FBNet | $\mathbf{6.42 \pm 0.32}$ | $\mathbf{3.64 \pm 0.16}$ | $\mathbf{4.02 \pm 0.21}$ | **45.17** | **79.25** | **69.66** |

Table B: Quantitative results of novel-view synthesis under the FID and IS metrics on the NAB and DOG datasets. $\uparrow$ indicates that higher is better, and $\downarrow$ indicates that lower is better. FBNet consistently outperforms the baselines.

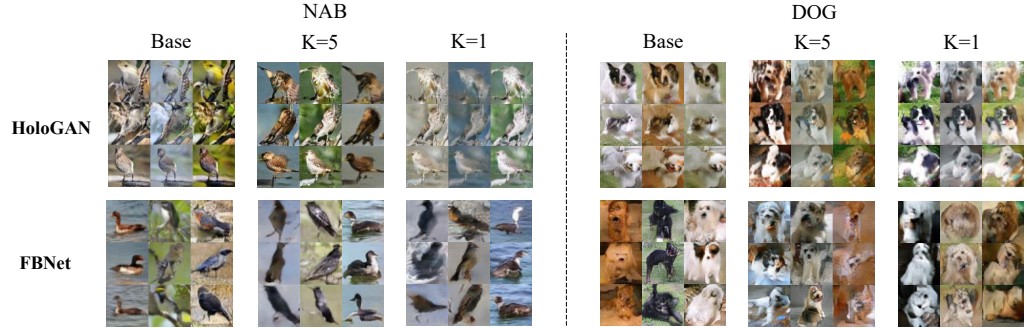

Figure D: Qualitative comparison of synthesized images from multiple viewpoints between our FBNet and state-of-the-art HoloGAN on the NAB and DOG datasets. Images in the same row/column are from the same viewpoint/object. The overall quality of the synthesized images for both methods indicates the general difficulty of the task, due to weak supervision and lack of data. However, our FBNet still captures the shape and attributes well, even though the data is scarcely limited. This shows the strong adaptability of FBNet for few-shot learning.

We randomly select 350 classes as base classes and 255 classes as novel classes. For the DOG dataset, there are 20,580 images belonging to 120 classes. We randomly select 80 classes as base and 40 classes as novel. For each class of the two datasets, we randomly select 75% as training data and 25% as test data. We pre-process these two datasets in a similar way as the CUB dataset.

Slightly different from the evaluation in the main paper, we only compare our **FBNet** with **Holo-GAN** (Nguyen-Phuoc et al., 2019) for the view synthesis task. For the recognition task, we compare our method with **FBNet-rec** and **FBNet-aug**. All the experimental setting remains the same as that in the main paper on the CUB and CompCars datasets.

**Quantitative Results:** Table A shows the recognition performance of the three competing models. Again, our method achieves the best performance for both base and novel classes. Table B shows the results of view synthesis and our method also achieves the best performance.

**Qualitative Results:** Figure D shows the synthesized images by HoloGAN and our FBNet on the two datasets. First, we note that the overall quality of the synthesized images for both methods becomes substantially worse than one would expect with large amounts of training images. This demonstrates the general difficulty of the task due to weak supervision and lack of data, indicating the need for the community to focus on such problems. Second, our FBNet significantly outperforms the state-of-the-art HoloGAN, especially for the diversity of the synthesized images. Additionally, even though the data is scarcely limited, FBNet still captures the shape and some detailed attributes of images well.

| Setting | PN | RN | MN | PMN | PN w/ G | MetaGAN | FBNet |
|---------|------|------|------|------|---------|---------|-------|
| Base | 46.05 | 45.89 | 46.72 | 47.10 | 48.57 | 39.91 | **49.63** |
| Novel-$K$=1 | 20.83 | 22.14 | 21.62 | 22.78 | 22.71 | 18.59 | **23.28** |
| Novel-$K = 5$ | 50.52 | 50.26 | 50.59 | 51.07 | 52.65 | 44.20 | **53.12** |

Table C: Top-1 (%) recognition accuracy for our approach and other state-of-the-art few-shot methods, including data hallucination-based methods on CompCars. Our FBNet consistently outperforms the other models.

|  | Model | Base | Novel-$K$=1 | Novel-$K$=5 |
|--|-------|------|-----------|-----------|
| CUB | PN (Snell et al., 2017) | 57.91 | 47.53 | 71.26 |
|  | FBNet-PN | **59.43** | 48.39 | **72.76** |
|  | RN (Sung et al., 2018) | 58.10 | 47.77 | 70.94 |
|  | FBNet-RN | 59.19 | **48.46** | 72.60 |
| CompCars | PN (Snell et al., 2017) | 46.05 | 20.83 | 50.52 |
|  | FBNet-PN | **49.63** | 23.28 | **53.12** |
|  | RN (Sung et al., 2018) | 45.89 | 22.14 | 50.26 |
|  | FBNet-RN | 48.99 | **24.83** | 52.72 |

Table D: Top-1 (%) recognition accuracy for recognition modules with PN (prototypical network) and RN (Relation Network), respectively, on the CUB and CompCars datasets. For base classes: **150-way** classification on CUB and **240-way** classification on CompCars; for $K$-shot novel classes: **50-way** classification on CUB and **120-way** classification on CompCars. The proposed FBNet consistently improves the performance of the single recognition modules for both PN and RN, indicating the generality of our framework.

## E    COMPARISON WITH OTHER FEW-SHOT RECOGNITION METHODS

We further compare our FBNet with a variety of state-of-the-art few-shot recognition methods on the CompCars dataset (Yang et al., 2015), including prototypical network (PN) (Snell et al., 2017), relation network (RN) (Sung et al., 2018), matching network (MN) (Vinyals et al., 2016), and proto-matching network (PMN) (Wang et al., 2018). We also compare with two data hallucination-based methods: PN w/ G (Wang et al., 2018) and MetaGAN (Zhang et al., 2018). Table C shows that our FBNet consistently outperforms these methods. Importantly, note that these methods can only address few-shot recognition, while our FBNet is able to deal with the joint task.

## F    ADDITIONAL EXPERIMENTAL RESULTS WITH RELATION NETWORK

The proposed feedback-based framework has a great generalization capability. In the main paper, we have shown that it is flexible with different backbone feature extraction networks $F$. In addition, our FBNet could improve the performance of the recognition module with different classification networks $P$ through feedback connections. To show this, we change the prototypical network (Snell et al., 2017) to the relation network (Sung et al., 2018) and report the recognition result on the CUB (Welinder et al., 2010) and CompCars (Yang et al., 2015) dataset in Table D. The experimental setting remains the same as that in the main paper. From the table, we can see that the whole feedback-based models consistently outperform the single recognition modules for the prototypical network and the relation network. This indicates that our proposed bowtie architecture is a general and robust framework that could improve the performance of different types of classification networks.

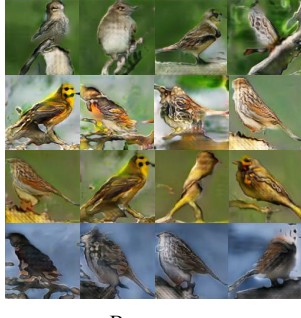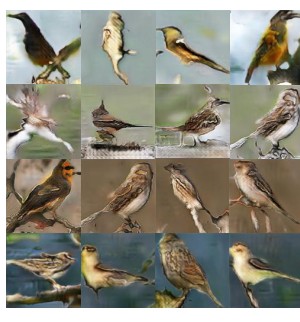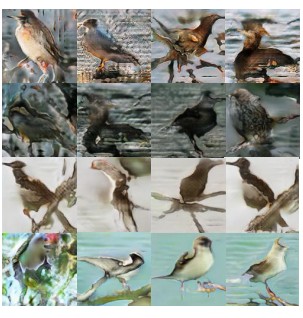

Base                                         K=5                                         K=1

Figure E: Synthesized 128-resolution images by our FBNet on the CUB dataset.

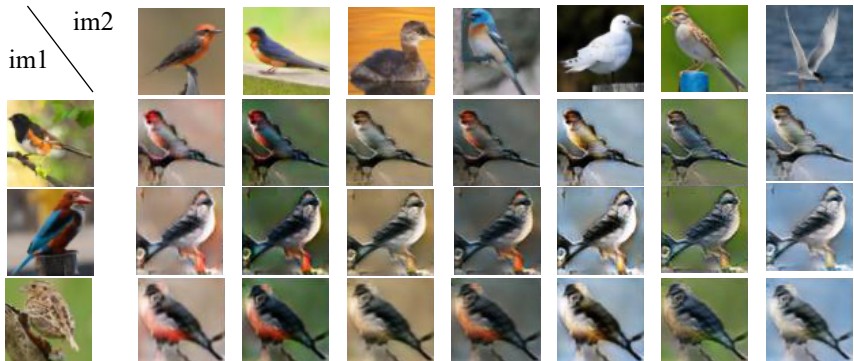

Figure F: FBNet with an extended attribute transfer task. 'im1' is the source image corresponding with 3D AdaIN and 'im2' is the attribute image corresponding with 2D AdaIN. Images in the same row/column have the same identity/attributes.

## G    EXPERIMENTS WITH HIGHER-RESOLUTION IMAGES

In the main paper, following the state-of-the-art novel-view synthesis model HoloGAN (Nguyen-Phuoc et al., 2019), we mainly focused on images of size $64 \times 64$. Our FBNet can also operate on higher-resolutions, by adding more 2D ResBlocks for the view synthesis module, and adding more convolution layers for the recognition module. Here, we modify our model to operate on images of size $128 \times 128$ and show some representative generated images on the CUB dataset in Figure E. We see that the proposed method effectively works with 128-resolution. We also note that higher resolution requires higher-quality training data. The unsatisfactory size and quality of the CUB dataset introduce additional challenges for synthesis, such as missing of details, inconsistency of identity across different viewpoints, and noisy background. However, such problems could be further addressed by improving the quantity and quality of the training data.

## H    FROM JOINT-TASK TO TRIPLE-TASK: ATTRIBUTE TRANSFER

The proposed bowtie framework could also be extended to address more than two tasks, by either introducing additional feedback connections or changing the architectures of the two modules. Here as an example, we introduce an additional "attribute transfer" task combined with the novel-view synthesis task. That is, instead of seeing one image each time, the view synthesis module sees one source image ('im1') and one target image ('im2') at the same time; then it generates images with the object of the source image and the attributes of the target image. We achieve this by arranging source latent input for 3D AdaIN units and target latent input for 2D AdaIN units in the view-synthesis module. Figure F shows the result of this additional task: images in the same row keep the same identity with im1; images in the same column have the same attributes of im2. Note that, to have a better visualization, we use the predicted views from the source images as the view input for all the generated images in Figure F, but the model could still synthesize images of different viewpoints.

