# OpenReview forum: "Bowtie Networks: Generative Modeling for Joint Few-Shot Recognition and Novel-View Synthesis"
_ICLR.cc/2021/Conference — ICLR 2021 Poster_

### Official Review · AnonReviewer3 · 2020-10-28
**Official Blind Review #3**

**Rating:** 6
**Confidence:** 3

**Review:**

This paper presents a new dual-task of joint few-shot recognition and novel synthesis. The main idea of this paper is to learn a shared generative model across the dual-task to boost the performances of both tasks. To achieve this, bowtie networks are employed to jointly learn geometric and semantic representations with a feedback loop. The proposed method is evaluated on fine-grained recognition datasets.

Pros:

1. This paper is well-written and easy-to-follow.
2. The idea of jointly learning the generative model and the proposed feedback loop is well-motivated.
3. The experiments on evaluating the view synthesis module is comprehensive and the results are promising.

Cons:

1. I think the contribution of this paper on the technical side is somehow weak. The recognition model and view synthesis modules are both adapted from existing works. The major contribution of this paper should be the bowtie architecture with the feedback loop, but this has also been well-studied in the literature.

2. During the evaluation of the recognition model, the proposed method should also be compared with other state-of-the-art methods on the same datasets other than only the variants of the proposed method. Although the authors claim that this is not the major goal of this paper, I think this is vital since one major benefit claimed by the author is that jointly learning the generative model can improve both tasks.

3. In Table 3, it is interesting to see that there is a performance gap when using simple (ResNet-18) and deeper (ResNet-50) models. Are there any intuitive explanations for this gap?

Overall, I like the main idea of the proposed method which learns a shared generative model across the dual-task. However, there are some concerns about the technical contributions and experiments. I will be happy to increase my rating if these concerns can be addressed in the rebuttal period.

---
Update after author feedback: I thank the authors for their reply. The authors have addressed all of my concerns. Therefore, I increased my final rating.

---

> ### Author Response · Authors · 2020-11-21
> **About the related work**
>
> We thank the reviewer for the comments. Regarding the comment “the major contribution of this paper should be the bowtie architecture with the feedback loop, but this has also been well-studied in the literature,” we are working on revising the related work to further discuss the difference between our approach and previous work. We were wondering if the reviewer could clarify which reference you are concerned about.

---

> > ### Comment · AnonReviewer3 · 2020-11-21
> > **related work**
> >
> > Thanks for the response. I would like to see a more detailed discussion on related works that jointly learn the data generation and task model. Some related works I am aware of include:
> >
> > [1] Jointly optimize data augmentation and network training: Adversarial data augmentation in human pose estimation, CVPR 2018.
> > [2] Learn to Augment: Joint Data Augmentation and Network Optimization for Text Recognition, CVPR 2020.
> > [3] A Unified Deep Model for Joint Facial Expression Recognition, Face Synthesis, and Face Alignment, TIP 2020.

---

> ### Author Response · Authors · 2020-11-25
> **Response to AnonReviewer3**
>
> We thank the reviewer for the valuable comments. The comments focus mostly on the discussion and comparison with related work. We address all the points as follows.
> 1.  Difference with related works that jointly learn the data generation and task model:
>
> We thank the reviewer for pointing out these related works. We have cited and discussed them in a new paragraph “Joint Data Augmentation and Task Model Learning” in the related work section of the revised submission.
>
> Existing works on jointly learning the data generation and task model only focus on the performance of the task model. By contrast, our proposed feedback-based bowtie network is a *bi-directional* feedback-based network. We focus on the dual-task of joint image synthesis and recognition. Therefore, the whole system contains two complete sub-networks, and the output of each module is fed into the other as one of the inputs to improve both. In addition, we address the generalization of the learned models from base to unseen novel categories, while they do not.
>
> Specifically, we are different from the mentioned references in important ways.
> [1] trains a generative data augmentation network and a discriminative pose estimation network jointly through adversarial loss. It aims to make the generative network to perform hard augmentation so that it can better facilitate the downstream pose estimation task. The augmentation only considers operations of scaling, rotating, and occluding, while we focus on a significantly more challenging task of synthesizing images from different viewpoints.
> [2] designs a controllable data augmentation method specifically for training a robust text recognizer, which is achieved by tracking and refining the moving state of the control points through feedback. The augmentation operation is highly specific to text images.
> Importantly, the above two papers mainly use adversarial training to optimize the target discriminative task but do not focus on the performance of the generative network. In comparison, our work focuses on the challenge *dual-task* of few-shot recognition and novel-view synthesis. We also use the bi-directional feedback connections while these two papers use single-directional (adversarial) feedback.
> [3] studies and makes use of the relationship among facial expression recognition, face alignment, and face synthesis tasks. Specifically, each task provides extra information for the other two tasks to have better training. However, the overall architecture of this work is a feedforward network rather than a feedback-based network. In comparison, the proposed FBNet uses bi-directional feedback to optimize the whole system.
>
> 2. Comparison with other state-of-the-art methods:
>
> We conduct additional experiments on the CompCars dataset and compare with a variety of SOTA few-shot methods, including prototypical network (PN), matching network (MN), relation network (RN), and proto-matching network (PMN). Importantly, we also compare with two data hallucination-based methods, including [4] and [5]. The results are summarized in Table 5 of the revised submission. Our FBNet consistently outperforms these SOTA baselines.
>
> 3. Any intuitive explanations for the gap when using simple (ResNet-18) and deeper (ResNet-50) models in Table 3:
>
> Considering that the total number of training images is, for example, 250 for 5-shot training and 50 for 1-shot training on the CUB datast, deeper models are easier to overfit in such few-shot learning scenarios. Deeper models generally work well for large-scale datasets and shallower models have a better performance with small-scale datasets. This phenomenon is also consistent with that in the few-shot learning literature, such as in the appendix of [6].
>
> [4] Wang Yu-Xiong, Ross Girshick, Martial Hebert, and Bharath Hariharan. "Low-shot learning from imaginary data." CVPR. 2018.
>
> [5] Zhang Ruixiang, Tong Che, Zoubin Ghahramani, Yoshua Bengio, and Yangqiu Song. "Metagan: an adversarial approach to few-shot learning." NeurIPS 2018.
>
> [6] Chen Wei-Yu, Yen-Cheng Liu, Zsolt Kira, Yu-Chiang Frank Wang, and Jia-Bin Huang."A closer look at few-shot classification."ICLR 2019.

---

### Official Review · AnonReviewer1 · 2020-10-28
**Interesting work, but missing important citation**

**Rating:** 6
**Confidence:** 5

**Review:**

This paper proposes a "feedback-based bowtie network" FBNet for joint generative synthesis via a GAN-based framework (specifically HoloGAN) and few-shot fine-grained recognition. The key idea of this work is to supervise both networks jointly via feedback mechanisms between the two, which helps to improve both tasks: image synthesis and few-shot recognition. The authors propose to use the synthesis network for synthesizing augmented images and additional losses computed by the image classification network along with conditional generation to improve the quality of the synthesized images.

Pros:
The authors consider a new setting of coupling HoloGAN with a downstream image recognition network and jointly training the two together. They show improvements both in image synthesis quality and image recognition by their approach. The experimental section is fairly thorough with many analyses and results presented.

Cons:
1. The proposed work's idea of coupling the synthesis HoloGAN network with a downstream visual learning task though a feedback mechanism between the two is not novel. This approach was previously introduced in the work Mustikovela et al., "Self-Supervised Viewpoint Learning From Image Collections", CVPR 2020, which bears much resemblance in approach to the current work, but trains for a different downstream task of viewpoint estimation versus few-shot categorial classifications. Mustikovela et al. also employ a conditional synthesis and various similar task-specific losses to jointly supervise both networks. The authors of this work should clearly cite this prior work and reframe the novelty of their approach in relation to it.

2. The paper lacks comparisons to existing SOTA few-shot learning techniques that employ strategies for hallucinating additional data/features for the training classes in the few-shot training settings, e.g., Wang et al., 2018 and Zhang et al., 2018. Is the proposed approach better than the previous few-shot learning approached that hallucinate data per class?

3. What dataset was the high resolution recognition network trained on?

------------
Post Rebuttal:
I thank the authors for their response. I am mostly satisfied with the authors' response to my (and other reviewers') concerns about properly citing prior works that jointly consider coupled image generation with downstream tasks and reframing the novelty of their work in juxtaposition to them. I would like to point out, however, that the authors' statement in the rebuttal "(2) we achieve bi-directional feedback while this work only implements the feedback from viewpoint estimation task to the generative network." is technically incorrect. The viewpoint estimation network in Mustikovela et al. is directly trained with images generated by the synthesis network under various viewpoints and hence it also achieves bi-directional feedback much like this current work. The authors should clearly re-frame their novelty and make this correction in the final version if accepted.

Nevertheless, I do feel that this work adds to the body of literature on joint conditional synthesis coupled with downstream vision tasks by (a) showing improvements in the quality of image synthesis achieved by considering downstream tasks and (b) by showing improvements in few-shot learning versus prior methods where only features are hallucinated, and (c) considering other applications beyond viewpoint estimation. Hence its contribution is above the acceptance threshold. I will maintain my previous rating.

---

> ### Author Response · Authors · 2020-11-25
> **Response to AnonReviewer1**
>
> We thank the reviewer for the valuable comments. The comments focus mostly on the discussion and comparison with related work. We address all the points as follows.
> 1. Difference with Mustikovela et al., "Self-Supervised Viewpoint Learning From Image Collections":
>
> We thank the reviewer for pointing out this related work. We have cited and discussed it in a new paragraph “Feedback-based Architectures” in the related work section of the revised submission.
>
> Specifically, different from existing feedback-based networks, our proposed feedback-based bowtie network is a *bi-directional* feedback-based network. We focus on the dual-task of joint view synthesis and recognition. Therefore, the whole system contains two complete sub-networks, and the output of each module is fed into the other as one of the inputs to benefit both.
>
> The work of Mustikovela et al. also leverages a generative model to facilitate another visual task (viewpoint estimation) through feedback-based training. However, we are different from Mustikovela et al. in three important ways: (1) We focus on the *dual*-task of novel-view synthesis and few-shot recognition, while they only focus on the downstream viewpoint estimation task; (2) We achieve bi-directional feedback, while they only implement the feedback from the viewpoint estimation task to the generative network；(3) We address the generalization of the learned models from base to unseen novel categories, while they do not.
>
> 2. Comparison of other SOTA methods, especially with some hallucination-based methods, e.g., Wang et al., 2018 and Zhang et al., 2018:
>
> Following the suggestion of the reviewer, we conduct additional experiments on the CompCars dataset and compare with the two hallucination-based methods, as well as a variety of other SOTA few-shot methods including prototypical network (PN), matching network (MN), relation network (RN), and proto-matching network (PMN). The results are summarized in Table 5 of the revised submission. Our FBNet consistently outperforms these SOTA baselines and the two hallucination-based methods.
>
> 3. What dataset was the high-resolution recognition network trained on?:
>
> We used the same data set for training the high-resolution recognition network. We resized the original images (resolution is higher than 224) to 224 X 224 when training the high-resolution recognition network, and resized the same images to 64 X 64 when training the low-resolution recognition network.

---

### Official Review · AnonReviewer2 · 2020-10-29
**The novel-view synthesis quality is bad, so the contributions of this paper are weakened.**

**Rating:** 5
**Confidence:** 3

**Review:**

The authors propose a model for joint few-shot recognition and novel-view synthesis. As shown in Figure 2, the model consists of two modules: view synthesis module and recognition module. The authors claim that these modules can help each other to become better. The two modules are trained jointly on each task alternatively.

Pros:
1. As both the justifications and experiments show, these two modules can help each other to be better.
2. Given the few-shot setting, the help from another task is meaningful.
3. The writing is clear, and makes me easy to follow.

Cons:
1. The novel-view synthesis quality is bad, so it can barely used for any other purposes besides assisting few-shot recognition module. In other words, the few-shot recognition result should be the only final output, and the novel-view synthesis output should only be considered as an intermediate result. So I think that the position of novel-view synthesis should not be lifted as high as the few-shot recognition, and this work should be dedicated to the few-shot recognition with a narrowed scope from "joint few-shot recognition and novel-view synthesis" to "few-shot recognition".
2. Based on the above point, then I doubt whether it is necessary to synthesize the pixels of other views, because the pixel quality is bad. Synthesizing the intermediate feature maps should be more realistic in this case, because the pixels are mainly for human (reviewers), but the featuremaps are mainly for model (recognition module). Human does not make the final few-shot recognition result better, but the recognition module does.
3. The experiments only show the fine-grained recognition results, e.g., fine-grained recognition for birds or cars. Given a single category (bird or car), the view synthesis quality is so bad, so I doubt if this module can be used for general recognition task involving many categories simultaneously. In that case, the novel-view synthesis module might make no sense at all. If that's the case, the scope of this work should be further narrowed from "few-shot recognition" to "few-shot fine-grained recognition".

Based on above points, I suggest rejecting this paper.

---

> ### Author Response · Authors · 2020-11-25
> **Response to AnonReviewer2 (1/2)**
>
> We thank the reviewer for the valuable comments. The comments focus mostly on the visual quality of the synthesized images. Below, we first address this most pressing concern and then address all the other remaining points.
> 1. Novel-view synthesis quality is bad:
>
> We would like to first clarify that the setting of our novel-view synthesis task is quite challenging: the images are synthesized *for novel classes based on few training data without any 3D/pose supervision* (as discussed in the related work Section “novel-view synthesis”). So we would like to kindly point out to the reviewer that we could not expect that the quality of the synthesized images would match those generated by typical GAN methods based on large training data and rich supervision. In fact, most existing methods on novel-view synthesis heavily rely on pose supervision or 3D annotation, which are not applicable in our case.
>
> Second, we compared our method with HoloGAN, the state-of-the-art method that is applicable to our setting. We substantially outperformed HoloGAN in both quantitative metrics (FID and IS) and visual quality (we better captured the shape and attribute details, and also better maintained the identity of the objects).
>
> Moreover, our method is important building blocks for novel-view synthesis under weak supervision and few-shot settings. The visual quality of our synthesized images can get significantly improved, for example with more training data. On the CUB dataset, the synthesized images on base classes (with roughly 45 training images per class) are significantly better than those on novel classes (with 1 training image per class). The synthesized base images are full of details and also keep the 3D identity from different views. To further demonstrate this, in the revised submission (Figure 5 and the highlights in Section 4), we provide results on the celebA-HQ dataset [1], which consists of aligned human faces. While celebA-HQ does not provide pose annotation, the aligned faces mitigate the pose issue to some extent. We find that in this case both the visual quality and diversity of our synthesized images significantly improve.
>
> 2. this work should be dedicated to the few-shot recognition with a narrowed scope from "joint few-shot recognition and novel-view synthesis" to "few-shot recognition":
>
> While we thank the reviewer for the comment, we disagree with the reviewer that our scope should be narrowed down on few-shot recognition because the novel-view synthesis quality is not as high as typical GAN methods. As explained above, we are addressing a much more difficult problem --- our synthesis setting “few training data without any 3D/pose supervision” is substantially more challenging than most existing novel-view synthesis methods. In addition, our synthesis quality significantly improves, for example with more training data or on datasets with better aligned poses.
>
> Besides, as mentioned by AnonReviewer4, “this paper is a combination of two tasks, combining two prior works to one system. The original part is the research question on whether the two tasks should share a common feature space and whether the results improve by the network model”. Our results indeed answered this question that joint few-shot recognition and novel-view synthesis *benefit each other*.
>
> More importantly, we believe that our dual-task of novel-view synthesis and few-shot recognition is an important first step towards in-the-wild recognition and reconstruction, where we will encounter the difficulty of weak supervision and lack of data. As mentioned by AnonReviewer4, “this is definitely a step in the right direction and I believe there is an interested audience for this finding and it is likely that the construction inspires future work”.

---

> ### Author Response · Authors · 2020-11-25
> **Response to AnonReviewer2 (2/2)**
>
> 3. Synthesizing the intermediate feature maps should be more realistic in this case:
>
> Most of this concern has been addressed above, demonstrating the importance of our dual-task of few-shot recognition and novel-view synthesis and the necessity of synthesizing images.
>
> Here we further compare the performance on few-shot recognition between synthesizing images and features. The new set of experiments is included in Table 5 and the new paragraph “Comparison with Other Few-shot Recognition Methods” of the revised submission. We can see that image-level synthesis outperforms feature-level synthesis. And also, image-level synthesis cannot address our novel-view synthesis task.
>
> 4. The scope of this work should be further narrowed from "few-shot recognition" to "few-shot fine-grained recognition:
>
> We focused on fine-grained categories, because (1) fine-grained few-shot recognition is a more challenging task than general few-shot recognition tasks [4], and (2) state-of-the-art novel-view synthesis models still cannot address image generation with broad types of general images [2]. Note that, as mentioned above, our goal in this paper is not to propose the best novel-view synthesis approach, but to propose a general framework with feedback connections that enables joint few-shot recognition and novel-view synthesis to benefit each other. That is, with the development of techniques for novel-view synthesis, our framework could be potentially extended to deal with broader types of images. For example, the recent work [3] proposes a 3D loss to enable a conditional GAN to perform novel-view conditional image synthesis. This work could be potentially integrated into our framework to replace HoloGAN to perform novel-view synthesis for large-scale general types of images. Further investigation is an excellent direction for future research. We include the above discussion in a new paragraph "Discussion and Future Work" at the end of Section 4 of the revised submission.
>
> [1] Lee Cheng-Han, Ziwei Liu, Lingyun Wu, and Ping Luo."Maskgan: towards diverse and interactive facial image manipulation." PAMI, 2020.
>
> [2] Liu Ming-Yu, Xun Huang, Arun Mallya, Tero Karras, Timo Aila, Jaakko Lehtinen, and Jan Kautz. "Few-shot unsupervised image-to-image translation." ICCV, 2019.
>
> [3] Noguchi Atsuhiro, and Tatsuya Harada. "RGBD-GAN: unsupervised 3D representation learning from natural image datasets via RGBD image synthesis." ICLR, 2020.
>
> [4] Yang Ze, Tiange Luo, Dong Wang, Zhiqiang Hu, Jun Gao, and Liwei Wang. "Learning to navigate for fine-grained classification." ECCV, 2018

---

### Official Review · AnonReviewer4 · 2020-10-30
**Well written paper on recognition + view synthesis approach**

**Rating:** 7
**Confidence:** 4

**Review:**

Summary:
Well written paper with solid experiments on an extension of two prior works. This is likely of interest, good quality and I recommend to accept the paper at ICLR. There are no extensions that I would propose to include in this version.

Quality:
Good quality, well written paper, easy to follow and sufficiently detailed. Content is on-topic for ICLR and of interest to a general audience.

Clarity:
The paper is very well written, all details on the model, the training procedures, experiments are included. This paper is well polished and was easy to read and follow. The main assumptions are stated early on, problem definition is well stated, and the goal of the experiments are clearly stated before going into their discussion. The Appendix provides additional results and details. Nice paper to read, thanks for putting in the effort.

Originality:
This paper is a combination of two tasks, combining two prior works to one system. The original part is the research question on whether the two tasks should share a common feature space and whether the results improve by the network model. So this is an interesting paper, I would assume there is quite an audience that is interested in this topic. Without doubt the model is well constructed and trained, so there is also value in the construction.

I have not seen the task of (few shot) recognition and visual reconstruction seen so far. This paper is a good extension of HoloGAN and has some novel points.
- Conditional version of HoloGAN. This is a simple extension but useful and serves the purpose.
- Combination of view synthesis and recognition. The flow of the architecture is well explained and leads to empirical improvements over each task in separation. More architecture choices would be possible, an evaluation of different backbones is included but not of other network combinations.
- Experimental results are sufficient, on established dataset, there is no novelty in the application.

Significance:
For both tasks (view synthesis and reconstruction) there are stronger models. The authors claim that other models could be combined in their setup, I agree, but the empirical results are below state-of-the-art. But this is definitely a step in the right direction and I believe there is an interested audience for this finding and it is likely that the construction inspires future work. There are some extensions that would go beyond the paper, such as more challenging data, images with more than one object, and combination with even more vision tasks.



----
Update and final recommendation.
I still recommend acceptance of the submission. The paper is well written, results stand on its own and the numbers improve in the way described. In light of the missing comparisons to other works pointed out by the fellow reviewers I have lowered my score because I think better calibrates with the significance of the work. Combination of downstream tasks is not novel but this combination I have not seen and so even bearing similarity with other approaches the paper still stands on its own.
Thanks to the reviewers and authors for their responses.

---

> ### Author Response · Authors · 2020-11-25
> **Response to AnonReviewer4**
>
> We thank the reviewer for the positive comments --- our work is an important step in joint few-shot recognition and visual reconstruction, and “there is an interested audience for this finding and it is likely that the construction inspires future work”.
>
> To address the proposed dual-task, we introduce a general feedback-based bowtie architecture that can be used on top of different view synthesis models and recognition models. And we instantiated this framework with a state-of-the-art view synthesis model (HoloGAN) and a widely adopted few-shot recognition model (prototypical network). In Section E of the appendix, we showed its generalization ability by replacing prototypical network with relation network. We agree with the reviewer that “more architecture choices would be possible” and “for both tasks (view synthesis and reconstruction) there are stronger models”.
>
> In fact, we also considered this issue when we instantiated our framework. Note that the setting of our novel-view synthesis task is quite challenging, which is under weak supervision without any ground-truth 3D annotations and with limited training data. By contrast, most existing methods on novel-view synthesis heavily rely on pose supervision or 3D annotation, which are not applicable in our case. At the time of the submission, HoloGAN was the state-of-the-art model with publicly available code that is applicable to our setting. Therefore, we adopted HoloGAN and outperformed it.
>
> We found that some most recent work addresses novel-view synthesis *alone* in a setting similar to ours; they can be potentially incorporated into our framework. For example, BlockGAN [1] is an extension of HoloGAN; RGBD-GAN [2] enables a conditional GAN (e.g., Self-attention GAN [3] and DCGAN [4]) to perform novel-view conditional image synthesis through a 3D loss. Unfortunately, the code of these two approaches has not been released, and we cannot include their results in this revision. We leave this as interesting future work, and we will include the integration of our framework with BlockGAN or RGBD-GAN in the next revision. We also included the above discussion in the paragraph "Discussion and Future Work" at the end of Section 4 in the revised submission.
>
> Regarding the reviewer’s comment on “combination with even more vision tasks”, we showed some results of extending our approach to address an additional “attribute transfer” task in Section G of the appendix. We agree with the reviewer that additional extensions, such as “more challenging data, images with more than one object, and combination with even more vision tasks” are all interesting and exciting. Further investigation is an excellent direction for future research.
>
> [1] Nguyen-Phuoc Thu, Christian Richardt, Long Mai, Yong-Liang Yang, and Niloy Mitra. "BlockGAN: learning 3D object-aware scene representations from unlabelled images." ICML, 2020
>
> [2] Noguchi Atsuhiro, and Tatsuya Harada. "RGBD-GAN: Unsupervised 3D representation learning from natural image datasets via RGBD image synthesis." ICLR, 2020
>
> [3] Zhang Han, Ian Goodfellow, Dimitris Metaxas, and Augustus Odena.  "Self-attention generative adversarial networks." ICML, 2019.
>
> [4] Radford Alec, Luke Metz, and Soumith Chintala. "Unsupervised representation learning with deep convolutional generative adversarial networks." ICML, 2016.

---

### Author Response · Authors · 2020-11-25
**General Response**

We thank all reviewers for their interest in our approach and their constructive and valuable comments.

We have uploaded a new version of the submission that takes into account the comments from all reviewers.
The main revision includes the following:
  1. We add two paragraphs in related work to discuss the difference between our work and (1) other feedback-based methods and (2) joint data augmentation and task model learning methods in detail.
  2. We add additional experimental results that compare with other state-of-the-art few-shot recognition models, including two data hallucination-based methods in Section 4 (Table 5).
  3. We visualize novel-view synthesis results on the celebA-HQ dataset in Section 4 (Figure 5).
  4. We add more analysis regarding Table 3.
  5. We add a paragraph “Discussion and Future Work” at the end of Section 4 to discuss the potential future research direction of this work.

The texts which we changed are highlighted as orange.

---

### Decision · Program_Chairs · 2021-01-07
**Final Decision**

**Decision:**

Accept (Poster)

**Comment:**

This paper uses an extension of HoloGAN for few shot recognition and novel view synthesis. All but one reviewer gave a final rating of accept. These reviewers were concerned that the submitted version of this work had not adequately placed this work in context with prior art. However, during the discussion these concerns seem to have been addressed sufficiently. The most negative reviewer was not impressed by the quality of the generated images; however these are relatively new methods and the few shot recognition aspect of this work is also part of the contribution. Accounting for all reviews and the discussion the AC recommends accepting this work as a poster.